# Meta-Regression Analysis of Relationships between Fibre Type and Meat Quality in Beef and Pork—Focus on Pork

**DOI:** 10.3390/foods12112215

**Published:** 2023-05-31

**Authors:** Michelle N. LeMaster, Robyn D. Warner, Surinder S. Chauhan, Darryl N. D’Souza, Frank R. Dunshea

**Affiliations:** 1School of Agriculture, Food and Ecosystem Sciences, The University of Melbourne, Parkville, VIC 3010, Australia; mlemaster@student.unimelb.edu.au (M.N.L.); ss.chauhan@unimelb.edu.au (S.S.C.); fdunshea@unimelb.edu.au (F.R.D.); 2SunPork Group, Brisbane, QLD 4009, Australia; darryl.dsouza@sunporkfarms.com.au; 3Faculty of Biological Sciences, University of Leeds, Leeds LS2 9JT, UK

**Keywords:** fibre type, tenderness, pork quality, WBSF, muscle, beef

## Abstract

This meta-regression analysis was conducted to identify the relationship between fibretype cross-sectional area (CSA) and frequency (%) and meat quality traits, especially tenderness (sensory and Warner-Bratzler Shear Force, WBSF). Literature searches were conducted using specific keywords which resulted in 32 peer-reviewed manuscripts that contained averages and correlation coefficients for fibre type (frequency and CSA) and quality traits of *longissimus* muscle for beef and pork (7 and 25 studies respectively). Correlations were analysed in meta-regression using R-Studio and linear regression was also conducted. For the combined beef and pork analysis, only pH, WBSF, and drip loss were associated with fibre type frequency and CSA (*p* < 0.05 for all). Limiting the analysis to pork, the key results were frequency of type I fibres were associated with decreased drip loss, increased cook loss, decreased lightness (L*) and increased sensory tenderness whereas frequency of type IIb fibres were associated with increased drip loss (*p* < 0.05 for all). In addition, the CSA of type I and IIb fibres was associated with colour traits lightness and redness (*p* < 0.05 for all). Future research should focus on fibre type across breeds and muscles to further understand the impacts of fibre type frequency and CSA on quality.

## 1. Introduction

Individual muscle fibre types are important determinants of overall meat quality. Specifically, the cross-sectional area (CSA) and frequency (%) of muscle fibre types affect muscle metabolism, colour, water-holding capacity (WHC, such as drip loss and cook loss), and both objective (Warner Bratzler Shear Force—WBSF) and subjective (sensory) tenderness [1,2].

There are three well-studied main fibre type isoforms: type I, type IIa, and type IIb, although other forms exist [3]. Fibre type composition impacts the rate and extent of postmortem pH decline due to variations in muscle metabolism [4,5,6]. Muscles composed of a greater percentage of type IIb fibres exhibit a faster or extended pH decline compared to muscles with a greater percentage of type I fibres [7,8,9,10]. Muscle fibre types also impact meat colour. For instance, a greater concentration of myoglobin in type I fibres yields a redder muscle compared to muscles composed primarily of type IIb fibres. Meat colour can also change due to variations in initial and ultimate pH, which are both impacted by fibre types. Muscles with predominantly type IIb fibres tend to be fast in pH fall, with a low initial pH; hence these muscles often have a pale colour associated with the pale, soft, exudative (PSE) condition [11]. In beef, glycolytic fibres are related to improved tenderness and ageing in cattle [12]. Conversely, muscles with predominantly type I fibres generally have lower muscle glycogen, and therefore often have higher pH and will appear dark, associated with the dark, firm, dry (DFD) condition [11,13,14]

Muscle fibre CSA and frequency can also impact WHC, including cook loss (%) and drip loss (%). The composition of fibre types in a muscle impacts the ability of fibre types to maintain WHC. Water distribution varies due to protein denaturation post-mortem and water loss during cooking which is related to the composition of different fibre types [15,16]. Specifically, muscles with predominantly type I fibres have a greater ability to maintain water-holding capacity due to the higher ultimate pH [17]. Conversely, muscles with a predominance of type IIb fibres generally have low WHC due to the inherently fast pH fall post-mortem as well as the lower ultimate pH [18]. When water distribution in the muscle changes, meat quality is impacted, including tenderness.

Tenderness is the main driving factor for repeated consumer purchases and is an important focus for the meat industry [19]. Fibre type CSA and frequency are known to impact WBSF and sensory tenderness [20]. Evaluating the relationships between CSA and frequency of fibre type isoforms and tenderness is important to understand changes in quality. Specifically, it is imperative to understand how the frequency and CSA of fibre types impact protein denaturation during cooking and subsequent impact on tenderness [21]. Muscles with more type I fibres generally have a higher denaturation temperature [22], delaying transverse and longitudinal shrinkage during heating, which likely explains the increased tenderness, compared to muscles with a greater composition of type IIb fibres [22]. The beef *masseter* is composed predominantly of type I fibres and has higher denaturation temperatures compared to the predominantly glycolytic beef *cutaneous trunci* [17]. However, studies within and between species have shown conflicting evidence on which fibre type isoform positively impacts tenderness. For instance, tenderness decreased in beef *semitendinosus* when moving in a transverse direction from superficial to deep layers, which was associated with an increase in type I fibres [23]. Conversely in beef *longissimus*, tenderness increased with the increased composition of type I fibres [24].

A deeper understanding of muscle fibre types in meat products and the correlation to eating quality is important to maintain and improve meat quality, in beef, and especially in the pork industry which has faced a rise of inconsistent products [25,26]. Established quantitative relationships between muscle fibre type frequency and CSA, and meat quality, lack consistency between studies. This regression analysis of published peer-reviewed data was conducted to identify the relationship between fibre type CSA and frequency and pork and beef quality. A deeper investigation was conducted specifically between pork muscle fibre types and overall quality parameters.

## 2. Materials and Methods

### 2.1. Literature Search and Data Collection

An extensive search of published peer-reviewed studies on the relationship between both cross-sectional area (CSA) and frequency (%) of fibre type and meat quality traits, was conducted using PubMed, Google Scholar, and ScienceDirect databases and utilizing the PRISMA protocol [6]. Additional studies were found in reference lists of relevant review papers. The dataset included 32 peer-reviewed full-text manuscripts (7 beef, 25 pork) published between 1985 and 2018 across journals, including the Journal of Animal Science, Meat Science, the Journal of Food Science, and the Journal of Food and Bioprocess Technology. Keywords used for searches included fibre type (cross-sectional area CSA or frequency %), pH, quality, tenderness, WBSF, cook loss, and drip loss in combination with meat or pork.

### 2.2. Manuscript Inclusion/Exclusion Criteria

For manuscripts to be included in this analysis, the following criteria were required: fibre type (CSA and/or frequency), species, muscles investigated, and tenderness (sensory and/or WBSF). Each study was required to give details of fibre type methodology and outline in detail the methodology used for the quality parameters, including colour, WHC, tenderness, and cook loss. Papers were rejected if these criteria were not met. Manuscripts analysed in the meta-analysis were required to provide the correlation coefficient (*r*). Papers that did not provide correlation coefficients but provided the averaged means were analysed in a random effect, simple linear regression. The details of the references used for the meta-regression analysis are given in Appendix A.

### 2.3. Database 

The author name, journal volume, date of publication, species (and breed when available), muscle fibre CSA or frequency were recorded in the database. The muscles analysed across the studies were constrained to the *longissimus.*

### 2.4. Data Analysis 

The meta-analysis was conducted based on previously established procedures [27] using RStudio (v2021.09.0 + 351.pro6 Ghost Orchid). Initially, the meta-regression analysis combined studies across species, including beef and pigs. Forest plots (Figure 1 and Figure 2) were created using RStudio (v2021.09.0 + 351.pro6 Ghost Orchid), where the correlation coefficient (*r*) was the fixed effect and the study was the random effect. The correlation coefficients tested were between fibre type I, IIa, IIb expressed as CSA or frequency, and pH, L*, a*, cook loss, drip loss, WBSF, and sensory tenderness. It is important to note that for all sensory tenderness analyses, the sensory tenderness scales were standardised across studies to a 1–10 scale where 10 indicated the highest sensory tenderness. Meta-analysis was conducted using the correlation coefficient (*r*) as the effect size. Weightings were calculated using the number of measurements as described by each study. The overall effect size estimates, 95% confidence intervals (95% CI), heterogeneity (I^2^) and *p*-value of heterogeneity (*p*) were reported. RStudio (v2021.09.0 + 351.pro6 Ghost Orchid) produces a default output of heterogeneity using the I^2^ statistic [28]. The I^2^ statistic describes the percent of the total variation of specific quality characteristics from the heterogeneity versus chance across all experiments. Heterogeneity can be classified into low (I^2^ < 49), moderate (I^2^ > 50–74) or high (I^2^ > 75) [29]. For ease of representation, the results of the forest plots from the combined species analysis are presented in Table 1, and pork-specific studies are presented in Table 2. Forest plots for the correlation of pork fibre types and drip loss and sensory tenderness are presented and include the author, overall effect, I^2^, and *p*-values. Appendix A is available with the list of the peer-reviewed manuscripts, author, year, and journal of publication.

For the linear regression of the pork-specific studies, the linear regression was between fibre type I, IIa, IIb (expressed as CSA or frequency) and pH, L*, a*, cook loss, drip loss, WBSF, and sensory tenderness. 

## 3. Results

Initially, both pork and beef studies were combined in an analysis for fibre type correlations with quality parameters (Table 1). The heterogeneity, or I^2^, for the regression analysis for the combined analysis of beef and pork studies, was moderate to high (moderate: I^2^ > 50; high: I^2^ > 75), indicating high variation between the beef and pork studies. Nevertheless, there was a positive correlation between the frequency of type I fibres and pH (I^2^ = 61%; 95% CI = 0.02, 0.16; *p* = 0.01) and a negative correlation to drip loss (I^2^ = 44%; 95% CI = −0.22, 0.03; *p* = 0.01). There was also a positive correlation between CSA of fibre type I and WBSF (I^2^ = 0%; 95% CI = 0.03, 0.13; *p* = 0.004). The frequency of fibre type IIa was positively correlated to WBSF (I^2^ = 0%; 95% CI = 0.04, 0.14; *p* = 0.0003). When pork and beef studies were analysed together, there was a lack of significance in L*, sensory tenderness, and IIb fibres for all quality parameters analysed. This lack of significance when pork and beef studies were combined is likely due to relatively high heterogeneity for many parameters possibly because of variations in muscle fibre composition between species. Therefore, a separate analysis was conducted including only the pork studies to investigate relationships between fibre types and quality parameters (Table 2). There were too few beef studies to analyse beef alone. Within the separate analysis of pork samples, there was a significant correlation between pH and fibre type IIa (I^2^ = 59%; 95% CI = 0.04, 0.24; *p* = 0.005). There was a significant correlation between the frequency of type I fibres and L* (I^2^ = 12; 95% CI = −0.22, −0.04; *p* = 0.05). The CSA of fibre type I (I^2^ = 0%; 95% CI =−0.2, −0.14; *p* < 0.001), IIa (I^2^ = 0%; 95% CI = −0.23, −0.09; *p* < 0.001), and IIb (I^2^ = 86%; 95% CI = 0.02, 0.28; *p* = 0.02) were correlated with L* where fibre type I and IIa were negatively correlated and fibre type IIb was positively correlated. The CSA of fibre type I was positively correlated with a* (I^2^ = 0%; 95% CI = 0.03, 0.15; *p* = 0.003) and type IIb was negatively correlated with a* (I^2^ = 82%; 95% CI = −0.32, −0.07; *p* = 0.004). The frequency of fibre type I was negatively correlated with drip loss (I^2^ = 0%; 95% CI = −0.27, −0.11; *p* = 0.004) and positively correlated with cook loss (I^2^ = 0%; 95% CI = 0.11, 0.21; *p* < 0.001). There was also a positive correlation between sensory tenderness and pork fibre type I (I^2^ = 0%; 95% CI = 0.05, 0.25; *p* = 0.001) and a negative correlation with type IIa (I^2^ = 41%; 95% CI = −0.26, −0.04; *p* = 0.006).

The simple linear regression from the pork studies analysed the average means from fibre type frequency and CSA across quality traits (Table 3). The linear regression across pork studies showed that the frequency (%) of pork fibre types I, IIa, and IIb were significantly correlated with meat pH. The frequency of fibre type I (*p* < 0.001) and fibre type IIa (*p* = 0.021) had a positive relationship with pH. The frequency of fibre type IIb (*p* < 0.001) had a negative relationship with pH. The frequency of fibre type IIa was negatively related to L* (*p* < 0.001) while the frequency of type IIb fibres was positively related to L* (*p* < 0.001). The CSA of fibre type I (*p* = 0.037) was negatively related to L*. The frequency of type IIa fibres was positively related to a* (*p* = 0.002) while the frequency of type IIb fibres was negatively related to a* (*p* < 0.001). The frequency of fibre type IIa was positively correlated with cook loss (*p* < 0.001) and negatively correlated with drip loss (*p* = 0.012). The CSA of fibre type IIb was positively correlated with cook loss (*p* = 0.027). The frequency of fibre type IIb had a positive relationship trend with drip loss (*p* = 0.053). The CSA of fibre type I was negatively correlated with WBSF (*p* = 0.014). Conversely, the CSA of fibre type IIb was positively correlated with WBSF (*p* = 0.009). The CSA of fibre type IIa had a positive relationship with sensory tenderness (*p* < 0.001). The CSA of fibre type I had a negative trend with sensory tenderness (*p* = 0.065) while the frequency of fibre type I was positively associated with sensory tenderness (*p* < 0.001).

## 4. Discussion

The main focus of this study was to investigate the influence of muscle fibre types on pork quality and, in particular, to identify if the frequency and/or CSA of fibre types improves overall meat quality. Pork muscle fibres and their impact on quality were investigated using both a meta-analysis and simple linear regression. Importantly, the results of the simple linear regression and meta-analysis concur and indicate the frequency of pork fibres had a more significant influence on pork quality than fibre type CSA. Specifically, the frequency of pork type Iib fibres had negative impacts on pork quality, including pH, colour, and water-holding capacity. Importantly, the frequency and CSA of fibre type I and Iia had a positive impact on both objectively measured tenderness and subjective sensory tenderness.

### 4.1. Variation within the Meta-Analysis across Species

Variations in the frequency and CSA of fibre types impact meat quality. When fibre types were analysed across species, the impact of fibre type CSA and frequency was underestimated probably due to species variation and differences in muscle metabolism between ruminants and non-ruminants [38,39]. There was a significant correlation between fibre type I and drip loss, where the presence of fibre type I reduced drip loss. However, there were limited significant effects of muscle fibre types on meat quality when analysing beef and pork studies combined. Therefore, because the primary focus of our research is the quality of pork, it was decided to analyse pork studies separately.

### 4.2. Meta-Analysis of Pork Only Studies—Water Holding Capacity

Fibre type isoforms are expected to impact overall meat quality. Importantly, the meta-analysis and the simple linear regression in this study had similar results where the effects of fibre type frequency were more significant for pH, colour, drip loss, cook loss, and sensory tenderness, compared to the impact of CSA on the same quality parameters. Similar findings have been reported in other studies indicating the frequency of muscle fibres is more important in determining quality rather than the cross-sectional area [30,31,38,40,41].

Water-holding capacity is an important parameter for meat quality. The frequency of fibre type I in this meta-analysis had a significant inverse relationship with drip loss. These findings are similar to previous studies by Kang et al. [30] who demonstrated a negative relationship between frequency of type I fibres and drip loss. The negative relationship between the frequency of fibre type I and drip loss is associated with improving water-holding capacity. Specifically, the results from this meta-analysis indicate that a greater frequency of fibre type IIb results in increased drip loss and an increased frequency of fibre type I would result in lower drip loss. Importantly, Choi et al. [42,43] also concluded that a greater frequency of fibre type Iib is associated with increased drip loss and more inconsistent water-holding capacity [8,44].

The frequency of fibre type I dem”nstr’ted a positive relationship with cook loss from this meta-analysis, contradicting other studies as well as the linear regression in this study. The discrepancies could be due to minimal studies in the meta-analysis comparing fibre type and cooking loss compared to a greater number of studies available for linear regression analysis. The linear regression indicated that there was no relationship between cook loss and the frequency of either fibre type I or iIb but there was a negative relationship between the frequency of fibre type iIa and cook loss. The linear regression also indicated that if the CSA of fibre type iIb increased, cook loss would increase which is supported by other studies [32] which found muscles with a greater frequency of oxidative fibres have improved water-holding capacity, compared to glycolytic fibres. Variations in fibre type frequency and CSA impact the ability of a muscle to retain water and maintain the structural integrity of the myofilament lattice framework [44].

### 4.3. Meta-Analysis across Pork Only Studies—Sensory Tenderness

Tenderness is considered the most important factor for eating quality [45,46]. Fibre types impact tenderness because different fibre types result in differing degrees of water-holding capacity and fibre shrinkage [47]. Using instruments to measure tenderness does not capture human perceptions and preferences for meat tenderness [48,49]. Therefore, it is important to include consumer sensory analyses [50]. Sensory tenderness scales were calibrated across all studies where an increase in tenderness scores is correlated to increased sensory tenderness, noting that higher sensory scores indicate greater tenderness whereas higher WBSF values indicate decreased tenderness. In the meta-analysis, fibre type I and IIa were related to sensory tenderness. Specifically, as the frequency of fibre type I increases, the sensory tenderness scores would increase indicating a more tender product. However, as the frequency of type IIa increases, the sensory tenderness scores decrease, indicating a tougher product. Other studies had similar results indicating oxidative fibres are positively related to tenderness [43,51]. The relationship between type IIa fibre frequency and sensory tenderness is similar to other data where a high frequency of type IIa fibres results in increased toughness [30]. Velotto et al. [33] concluded that type I fibres are positively related to sensory tenderness but also found significant negative relationships between type IIb fibres and sensory tenderness which was not observed in this meta-analysis. Other reviews have commented on the discrepancies between studies on the relationship between pork fibre types and physiochemical properties and the impact on overall pork quality [43,52]. It is important to note that several breeds were analysed within these pork studies which could account for differences on the relationship between fibre type and sensory tenderness, in agreement with Wojtysiak & Poltowicz [40], who found that pork quality was significantly different between traditional and commercial pigs. These results indicate that the frequency of fibre types has an important impact on sensory tenderness [1,2,53]. Importantly, pork muscles composed predominantly of type I fibres are often more tender while pork muscles composed predominantly of IIa and IIb are often tougher. Future research could focus on the impacts of muscles with different fibre type frequency to further investigate impacts on both objective and sensory tenderness.

### 4.4. Meta Analysis and Linear Regression of Pork Only Studies—WBSF

In the meta-analysis, there was no significant correlation between either frequency or CSA of the three fibre types and WBSF. Therefore, we investigated further using the simple linear regression which indicated a negative relationship between the CSA of fibre type I and WBSF and a positive relationship between the CSA of fibre type IIb and WBSF. Therefore, as the CSA of fibre type I increases, the WBSF decreases resulting in a more tender product and as the CSA of fibre type IIb increases, the WBSF increases, resulting in a tougher product. Jeong et al. [35] found that as fibre CSA, especially from more glycolytic fibres, increased, tenderness decreased. This is similar to studies for beef [52] and pork [53] where larger fibre sizes resulted in tougher meat. Pork loins are composed predominantly of type IIb fibres which could impact WBSF [30,32,54]. Muscles predominantly composed of type IIb fibres resulted in tougher products in studies by Maltin et al. [53] and Henckel [36]. Type IIb fibres also have reduced spacing between myofilaments but increased spacing between fibres which results in increased transverse shrinkage during cooking, increasing toughness [22,55]. Conversely, type I fibres have smaller CSA [32] and therefore exhibit reduced transverse shrinkage.

### 4.5. Pork Fibre Type and Colour

Fibre types impact the lightness and redness of pork. The meta-analysis indicated a significant correlation between the frequency of type I and L* where the increased frequency of type I fibres results in a lower L* or darker product. Importantly, the meta-analysis found that the CSA of all three fibre types significantly impacts L*. Specifically, the CSA of both type I and IIa fibres had a negative correlation with L* while the CSA of type IIb had a positive correlation with L*. Redness was also impacted by the CSA of fibre type I and IIb where there was a positive correlation between type I and a* and a negative correlation between type IIb and a* indicating that as type IIb CSA increases, the redness decreases which could result in pale products. Muscles with increased concentrations of fibre type I often yield a more desirable pinkish-red colour [56] due to a greater concentration of myoglobin which results in a lower L* and higher a* compared to muscles primarily composed of type IIb fibres [57]. The relationships between fibre type CSA and L* and a* indicated that increased type IIb fibres [2] are associated with lighter-coloured, less red products while increased type I fibres [34] are associated with darker, redder products. Colour is also impacted by postmortem pH due to variations in the rate of myoglobin oxidation. Low pH increases the rate of myoglobin denaturation and negatively impacts the oxidation-reduction reaction, destabilising myoglobin and negatively impacting colour. Muscle fibre CSA impacts the rate of light diffraction and scatters light, altering the colour [14]. A darker colour is associated with fibre swelling due to increased water retention which increases the myofilament lattice spacing and reduces light scattering, yielding a darker product [14,58]. Some studies have also shown that a greater number of total fibres restricts the size of the fibres which ultimately lowers drip loss [59].

### 4.6. Meta-Analysis across Pork Only Studies—pH

In this meta-analysis, there was a significant correlation between the frequency of fibre type IIa and pH indicating that as the frequency of fibre type IIa increased, the pH would increase. Interestingly, this meta-analysis did not demonstrate a significant correlation between type I or IIb fibres and pH. Therefore, we investigated these relationships further using linear regression. The linear regression showed that the frequency of type I and IIa fibres was positively related to pH, and the frequency of type I was negatively related to pH. These findings demonstrate the importance of the fibre type frequency within the muscle on postmortem pH due to the impact of fibre type on metabolic activity. Ryu & Kim [31] also found that increased percentages of fibre type IIb in pork resulted in lower ultimate tissue pH. The frequency of type I fibres is related to slower pH fall and a higher ultimate pH due to lower glycogen content in oxidative fibres [60]. Muscles that are composed predominantly of type IIb fibres, such as pork *longissimus*, can exhibit a rapid pH decline due to increased glycolytic activity and glycolysis postmortem [61] compared to muscles that are generally more oxidative, such as beef muscles in comparison to pork [62,63,64,65,66]. Fibre type composition can also impact the rate and extent of pH decline across different breeds [67]. For instance, pigs that are bred to produce leaner, heavier-muscled carcasses have a greater frequency of type IIb fibres and a faster pH decline, compared to breeds that yield fatter, lighter carcasses, such as heritage breeds, or breeds which have a greater composition of type I fibres [31]. There were multiple breeds included in this meta-analysis which could have impacted the significance of the results. Therefore, investigating the genetic impact of pork fibre types and overall quality between breeds could help determine a deeper and more fundamental understanding of how genetic changes impact specific muscle fibre type frequency and ultimately overall pork quality.

## 5. Conclusions

This meta-analysis was conducted to gain a more comprehensive understanding of the impacts of muscle fibre type CSA and frequency on beef and pork quality with a focus on pork. The meta-analysis indicated important relationships between pork fibre type and the quality traits pH, drip loss, cook loss, sensory tenderness and colour. Specifically, type I fibres in pork were associated with improved water-holding capacity, darker and redder meat and more tender meat, compared to type IIb fibres. This study indicates that pork water holding capacity and sensory tenderness are impacted more significantly by variations in fibre type frequency rather than variations in CSA.

## Figures and Tables

**Figure 1 foods-12-02215-f001:**
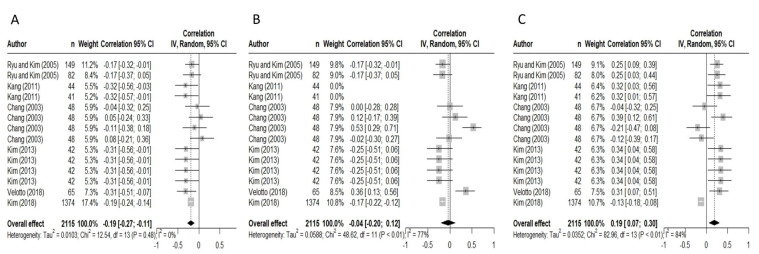
Forest Plots demonstrating the meta-regression for the correlations between drip loss (%) and pork fibre type frequency (%) I (**A**), IIa (**B**), and IIb (**C**) [2,30,31,32,33,34].

**Figure 2 foods-12-02215-f002:**
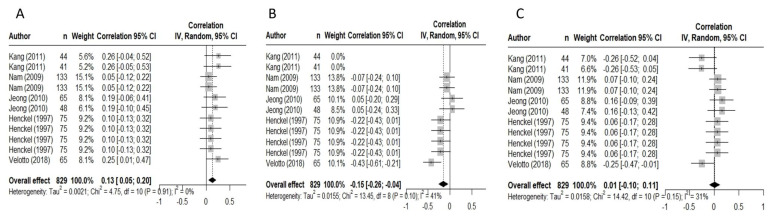
Forest Plots demonstrating the meta-regression results for the correlations between sensory tenderness and pork fibre type frequency (%) I (**A**), IIa (**B**), and IIb (**C**) [30,33,35,36,37].

**Table 1 foods-12-02215-t001:** Meta-regression analysis of the correlation coefficients for frequency of fibre types (%) and cross-sectional area (CSA) of I, IIa, and IIb and various quality parameters across both beef and pork studies. The heterogeneity (I^2^) and *p*-value are shown for the quality characteristics pH, lightness (L*), WBSF, drip loss (%), and sensory tenderness.

Quality Characteristic	Fibre Type	Overall Effect	d.f.	I^2^	95% CI
**pH**	I	0.09	37	61%	0.02; 0.16 *
**(% fibre type)**	IIa	0.03	35	61%	−0.05; 0.10
	IIb	−0.04	36	60%	−0.10; 0.03
**L***	I	−0.03	13	62%	−0.17; 0.10
**(% fibre type)**	IIa	−0.12	12	54%	−0.30; 0.06
	IIb	0.08	13	66%	−0.09; 0.25
**WBSF N**	I	−0.04	66	26%	−0.09; 0.02
**(% fibre type)**	IIa	0.10	64	0%	0.04; 0.14 *
	IIb	0.00	66	48%	−0.06; 0.07
**WBSF N**	I	0.08	61	0%	0.03; 0.13 *
**(CSA fibre type)**	IIa	0.00	59	0%	−0.05; 0.05
	IIb	0.02	59	49%	−0.05; 0.09
**Drip loss %**	I	−0.19	20	44%	−0.22; −0.03 *
**(% fibre type)**	IIa	−0.02	18	75%	−0.15; 0.10
	IIb	0.14	20	85%	−0.00; 0.27
**Sensory**	I	0.01	10	31%	−0.10; 0.11
**Tenderness**	IIa	−0.11	9	66%	−0.24; 0.03
**(% fibre type)**	IIb	−0.03	11	57%	−0.15; 0.08

* Significance determined by the 95% CI across all studies.

**Table 2 foods-12-02215-t002:** Meta-regression analysis of the correlation coefficients for frequency of fibre types (%) and cross-sectional area (CSA) of I, IIa, and IIb and various quality parameters for the pork studies only. The heterogeneity (I^2^) and *p*-value are shown for the quality characteristics pH, lightness (L*), WBSF, drip loss (%), and sensory tenderness.

Quality Characteristic (CSA or %)	Fibre Type	Overall Effect	d.f.	I^2^	95% CI
**pH**	I	0.09	18	67%	−0.00; 0.18
**(% fibre type)**	IIa	0.14	18	59%	0.04; 0.24 *
	IIb	−0.09	18	77%	−0.19; 0.01
**L***	I	−0.13	13	12%	−0.22; −0.04 *
**(% fibre type)**	IIa	−0.09	12	89%	−0.27; 0.11
	IIb	0.08	14	63%	−0.04; 0.19
**L***	I	−0.20	6	0%	−0.26; −0.14 *
**(CSA fibre type)**	IIa	−0.17	6	0%	−0.23; −0.09 *
	IIb	0.16	6	86%	0.02; 0.28 *
**a***	I	0.09	6	0%	0.03; 0.15 *
**(CSA fibre type)**	IIa	0.03	6	37%	−0.05; 0.11
	IIb	−0.21	6	82%	−0.32; −0.07 *
**WBSF N**	I	0.05	17	57%	−0.04; 0.14
**(% fibre type)**	IIa	0.06	15	0%	−0.01; 0.13
	IIb	−0.05	17	37%	−0.13; 0.04
**WBSF N**	I	0.07	16	58%	−0.04; 0.17
**(CSA fibre type)**	IIa	−0.02	14	0%	−0.09; 0.05
	IIb	−0.04	14	54%	−0.14; 0.07
**Drip loss %**	I	−0.19	10	0%	−0.27; −0.11 *
**(% fibre type)**	IIa	−0.04	11	77%	−0.20; 0.12
	IIb	0.19	13	84%	0.07; 0.30 *
**Cook loss %**	I	0.16	7	0%	0.11; 0.21 *
**(% fibre type)**	IIa	−0.17	5	97%	−0.59; 0.33
	IIb	−0.05	7	84%	−0.17; 0.06
**Sensory**	I	0.13	10	0%	0.05; 0.20 *
**Tenderness**	IIa	−0.15	8	41%	−0.26; −0.04 *
**(% fibre type)**	IIb	0.01	10	31%	−0.10; 0.11

* Significance determined by the 95% CI across pork studies.

**Table 3 foods-12-02215-t003:** The relationship between pork fibre types I, IIa, and IIb cross-sectional area (CSA) and frequency (%) and quality parameters pH, lightness (L*), redness (a*), cook loss (%), drip loss (%), Warner-Bratzler Shear Force (WBSF), and sensory tenderness using simple linear regression. The slope, number (*n*), standard error (SE) and *p*-value are shown.

	Cross-Sectional Area			Frequency		
Trait Fibre Type	*n*	Slope ± SE	*p*-Value	*n*	Slope ± SE	*p*-Value
**pH**						
I	25	0.000044 ± 0.0000522	0.41	51	0.019 ± 0.0035	<0.001
IIa	25	−0.000014 ± 0.0000455	0.76	51	0.0073 ± 0.00304	0.021
Iib	25	−0.000021 ± 0.0000256	0.43	51	−0.0090 ± 0.00194	<0.001
**L***						
I	15	−0.0028 ± 0.00122	0.037	42	−0.341 ± 0.192	0.084
Iia	15	−0.00102 ± 0.00133	0.84	42	−0.56 ± 0.134	<0.001
Iib	15	0.00031 ± 0.000705	0.670	42	0.399 ± 0.0859	<0.001
**a***						
I	19	0.00013 ± 0.000356	0.72	42	0.026 ± 0.206	0.90
Iia	19	0.00015 ± 0.000336	0.34	42	0.50 ± 0.147	0.002
Iib	19	0.00014 ± 0.000208	0.52	42	−0.38 ± 0.0903	<0.001
**Cook loss** %						
I	10	−0.0018 ± 0.00340	0.62	20	0.17 ± 0.627	0.79
Iia	10	−0.0094 ± 0.00865	0.31	20	2.55 ± 0.520	<0.001
Iib	10	0.0069 ± 0.00263	0.027	20	−0.88 ± 0.424	0.065
**Drip loss** %						
I	16	−0.0010 ± 0.000816	0.22	47	−0.065 ± 0.0594	0.28
Iia	16	−0.00035 ± 0.000806	0.67	47	−0.19 ± 0.0073	0.012
Iib	16	0.00030 ± 0.000426	0.50	47	0.099 ± 0.0488	0.053
**WBSF** N						
I	4	−0.0056 ± 0.00999	0.014	31	−0.0092 ± 0.0713	0.90
Iia	4	−0.0023 ± 0.00333	0.73	31	−0.016 ± 0.379	0.73
Iib	4	0.0022 ± 0.000601	0.009	31	0.0035 ± 0.0318	0.91
**Sensory tenderness**						
I	8.5	−0.00125 ± 0.000619	0.065	24	1.53 ± 0.842	0.10
Iia	13	0.0039 ± 0.00291	<0.001	24	−1.09 ± 0.362	0.008
Iib	13	−0.0013 ± 0.000557	0.78	24	−0.32 ± 0.926	0.74

## Data Availability

Data is contained within the article or Appendix A.

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
