# Peer review of "Meta-Regression Analysis of Relationships between Fibre Type and Meat Quality in Beef and Pork—Focus on Pork"

_foods, 2023, doi:10.3390/foods12112215_

Round 1
Reviewer 1 Report
The paper summarizes the knowledge on the impact and relationship of the cross-section area and muscle fiber type on the meat quality traits in beef and pork. In this regard, in my opinion, the title should be changed and "beef" should be included too. Later in the abstract/ introduction, when defining the objective of the meta-analysis, the authors might mention, that despite beef and pork has been assessed, the focus is on pork.
The abstract and the key words should also be corrected and the results concerning beef should also be included.
The Introduction as a whole contains the necessary information to convince of the necessity for doing this research, however, again, some world on beef should also be included. Otherwise, why have the authors done all the search and assessment of the literature on beef?
The material and method section should be improved. The table that is intended to present the studies should be more detailed. In my opinion, it should include information about the parameters each study has focused on, also, the number of samples/ animals included in each individual paper, breed, please be as more detailed as possible. Usually such tables are large but are necessary so that readers obtain clearer idea about the data set for the meta-analysis. Please, add a reference about the limits of the the heterogeneity (line 109-110).
Manuscript inclusion/ exclusion criteria of the papers should also contain the availability of any measure on intra-experiment variation, expressed either as standard error (SE) or standard deviation (SD) of the mean of each treatment group, or alternatively as mean square error (MSE) or root mean square error (RMSE).
It remains unclear how is the effect size was calculated? Using the raw mean difference or the standardized mean difference? Usually meta-regression is preformed when significant heterogeneity was detected, and covariates possible leading to this heterogeneity are tested ( for instance breeds, as authors have stated that numerous breed were included and might have impacted the results). However, this is not clearly described in the material and method section and should be improved.
In regard to results presentation, in my opinion, beef studies should also be illustrated using forest plots. Still remains unclear why the authors preformed meta-analysis on beef and pork together and reported the effects together in table 1. The discussion of the results on the pork studies is skillfully performed.
The conclusions should be revised and be more precise.
Line 350. "This meta-analysis was conducted to gain a more comprehensive understanding of the
impacts of muscle fibre type CSA and % on pork quality". But studies on beef were also included and presented.
Line 351-354. The meta-analysis indicated an important relationship between fibre type and the quality traits pH, drip loss, cook loss sensory tenderness and colour. Specifically, type I fibres had improved water holding capacity, darker and redder meat and mire tender meat, compared to type IIb fibres. Where ? In beef or pork?
Apart
Author Response
Thanks for your excellent comments - please see attached for our resposne to each comment.

Reviewer 2 Report
Dear authors,
Thank you for your great effort in finishing this manuscript. The manuscript with the title of “Meta regression analysis of relationships between fibre type and meat quality with a focus on pork” was seamlessly written.
Started with the introduction that provided detailed explanation of the investigated topics, logical thought, and presentation of research gap. With the intention to elaborate the relationship between muscle type and frequencies on textural and sensorial quality traits of pork and beef, I believe that this manuscript can contribute to the scientific significances of textural studies.
Materials and methods presented a detailed procedure on how this study was done and assure the reproducibility.
The discussion was also elaborated in an easy to be understood fashion, accompanied by a suitable result.
Sincerely,
Author Response
Thank you for your comments and your time reviewing our manuscript. We appreciate the time you spent on our manuscript.
Reviewer 3 Report
The article issue is original, well planned and presented. Trial plan and discussion are good.
Author Response

(The authors gave the same response as above.)

Round 2
Reviewer 1 Report
The authors have taken into account the remarks that I have made concerning the manuscript. It has now significantly improved and can be accepted and proceed to publication.